# Experimental and Numerical Evaluation of Mechanically Stabilized Earth Wall with Deformed Steel Bars Embedded in Tire Shred-Sand Mixture

**Beenish Jehan Khan** [1,*] **, Mahmood Ahmad** [2,3] **, Mohanad Muayad Sabri Sabri** [4,*] **, Irshad Ahmad** [5] **, Bakht Zamin** [1] **and Mariusz Niekurzak** [6]

1 Department of Civil Engineering, CECOS University of IT and Emerging Sciences, Peshawar 25000, Pakistan; bakht@cecos.edu.pk
2 Department of Civil Engineering, Faculty of Engineering, International Islamic University Malaysia, Jalan Gombak, Selangor 50728, Malaysia; ahmadm@iium.edu.my
3 Department of Civil Engineering, Bannu Campus, University of Engineering and Technology Peshawar, Bannu 28100, Pakistan
4 Peter the Great St. Petersburg Polytechnic University, 195251 St. Petersburg, Russia
5 Geotechnical Earthquake Engineering, Department of Civil Engineering, University of Engineering and Technology, Peshawar 25120, Pakistan; irspk@yahoo.com
6 Faculty of Management, AGH University of Science and Technology, 30-059 Krakow, Poland; mniekurz@zarz.agh.edu.pl
* Correspondence: beenish@cecos.edu.pk (B.J.K.); mohanad.m.sabri@gmail.com (M.M.S.S.)

**Abstract:** This article presents the experimental and numerical analysis behavior on Mechanically Stabilized Earth Wall (MSE) under applied overburden load performed on the 1.5 m high, 0.9 m width, and 1.2 m length reinforced with deformed steel bars embedded in sand alone and tires shred-sand mixture. The study investigates how deformed bars, strength, and geometry affect the failure mechanism. The top of the wall was laden with additional overburden weight at various stages to explore the pre-failure wall behavior. The horizontal displacements were measured using potentiometers of the wall face and by potentiometers placed at the top of the loading plate. The results of the observations were compared to the analysis results derived from a numerical model created using the Plaxis 3D software. Numerical modeling was also applied to assess the behavior of MSE wall (3D model) on the failure mechanism of the walls. The parameters for the numerical models were derived from independent tests results, which were compared with the experimental observations. A good level of agreement with measurements was confirmed for the 3D model with the experimental data. From the results, it was deduced that at 30 kPa and 40 kPa, the tire shred-sand mixture with reinforcement gave a 36% and 58% reduction in face deflection compared to sand with reinforcement. The difference between numerical and experimental values ranges from 12% to 15%.

**Keywords:** sand; deformed steel bar; tire shred; Plaxis 3D; mechanically stabilized earth wall





## 1. Introduction

Due to the increasing construction of the country's national infrastructure, mechanically stabilized earth walls (MSE) have become a preferred alternative for stabilizing slopes and ecological greening on limited land resources all over the world [1–4]. In 1966, a French architect named Henry Vidal was able to obtain a patent for what he called "reinforced earth." An MSE wall is a composite structure made up of soil reinforcement, backfill, and a face component [5–8]. The MSE wall is built by embedding the reinforcement in backfill material and layering it; this reinforced mass resists the earth pressure caused by the retained soil by employing relative motion between the reinforcement and the soil. As a result, the MSE wall's performance is determined by the interaction of the components, particularly the soil and reinforcement [8,9]. Steel strips, steel grids, and polymeric strips are the most

common soil-reinforcement elements. According to the Federal Highway Administration, the required reinforcement length for developing efficient soil-reinforcement interaction is 0.7H (H = height of the MSE wall) [3,6,9,10].

Backfill soil selection is important in addition to reinforcement length, as Soong and Koerner (1999) documented 20 case histories of geosynthetic-reinforced wall failures induced by the use of marginal backfill soil [11]. Backfill for MSE walls should be a free-draining material that is devoid of organic and deleterious material, according to American Association of State Highway and Transportation Officials (AASHTO) requirements [8–10]. Furthermore, the reinforced backfill material must withstand the earth's pressure exerted by the retained soil mass. The pressure exerted by the retained fill is resisted by the reinforced soil mass. As a result, the influence of retained fill during the design phase should be carefully studied [6,12,13].

The study of MSE walls has continued for decades in pursuit of creative designs that can compete with the global world's requirement for sturdier, economical, faster, and more aesthetic retaining walls [14]. Tire shards, sawdust, fly ash, plastic bottles, geofoam, and other materials are used for sand-based backfill and soil stabilization [15–21]. Raw earth is one of the oldest building materials, appropriate for a wide range of applications, including load-bearing walls, plasters, and finishes. The addition of straw fibers might cause this material to behave differently. Fibers would increase the ductility of earth elements, and the consequent mechanical production and compaction methods for the specimens [22]. The lightweight materials in retaining walls help to reduce vertical loads and lateral displacements [23,24]. When compared to the attributes of other materials used as backfill material, tire shreds offer distinct characteristics, such as being elastic and lightweight [25,26]. Furthermore, discarded tires are attractive for geotechnical applications due to their high thermal conductivity, low density, and high shear strengths at large strains [23,27,28]. Due to the rise of human populations and the increased usage of automobiles, millions of rings of scrap tires are removed from the consumption cycle each year and collected as rubbish. Due to the vast volume of scrap tires, the prevalence of dangerous fires, and the significant expenditure of hygiene disposal, this has become one of the most critical environmental concerns [29,30]. As a result, it is recommended that certain remedies to the issue are required. One approach is to reuse them as filler materials in building projects such as road construction, retaining walls, and drainage systems [31].

Over the last two decades, numerous experimental and field research on the use of tire shreds as a backfill material for MSE walls have been conducted. Tarek et al. (2004) investigated fully instrumented modular walls with soil-tire chips as a backfill material, and geotextiles and geogrid for reinforcement. The wall was loaded with a surcharge after construction and monitored to determine the mass behavior of soil-tire chip backfills. Earth pressure cells, position transducers to measure wall face displacement, and strain gauges to monitor strains in the geosynthetic reinforcement were installed on the wall. As the surcharge increased, the strain in the geotextile and geogrid layers also increased [32]. Based on field instrumentation and testing, Rodrigo et al. [33] examined the possibility of employing a mixture of tire shreds and soil as a fill material for embankments. The successful construction and functioning of tire shred embankments could encourage the use of tire shred as a fill material, which would have significant societal advantages. Settlement monitoring with settlement plates, vertical and horizontal inclinometer monitoring, temperature monitoring, and groundwater quality analysis are all part of the apparatus. The results revealed that vertical and horizontal settlement were both around 5 mm, with no differential settlement and no signs of slope stability issues, cracking on the road or erosion. In study [34], this research focused on two full-scale model tests on mechanically stabilized earth (MSE) walls. The first test was done with a stiff wall face, and the second test was done with a flexible wall face. The horizontal and vertical ground pressures were used to compute the loads and strains on the reinforcement, and the results were compared to analytical models. To examine the face and vertical deflection, as well as strains created in the reinforcement, various instrumentation was installed, including vertical and horizontal

cell pressure, load cells, strain gauges, and linearly variable displacement transducers (LVDT). According to the findings, flexible facing exerts a lower tensile strain on reinforcing layers than rigid facing. The reinforcing layers in the top strata, directly below the strip footing load, had the most stresses. The maximum wall deflection of the flexible facing is larger than that of the stiff face.

A previous study has shown that scrap tires can be used in geotechnical applications. Furthermore, despite all of the combined efforts, there is still a lack of understanding about using various types of deformed steel bars embedded in a scrap tire-sand mixture as backfill material. Furthermore, a review of the literature reveals that a substantial amount of research has been done to assess the resilience of geotextile materials.

However, more research into the behavior of deformed bars embedded in MSE walls is required. An instrumented large-scale model validated by numerical modeling of the mechanically stabilized earth wall has been used to investigate the behavior of deformed steel bars placed in tire shred-sand mixture. Based on the findings, it was deduced that the deformed steel bars embedded into the sand and tire shred mixture serve an essential function in MSE walls as reinforcement, enhancing the mechanical behavior of mechanically stabilized earth walls.

Despite the extensive theoretical and experimental studies, the utilization of MSE wall, advancements in present design and analysis procedures, and the adoption of new approaches must be validated against measurements from instrumented walls. However, the number of instrumented field walls is limited, and this number is substantially lower when candidate walls are limited to structures with appropriate high-quality data for component materials and evaluation of comprehensive wall performance [35,36]. Numerical models of reinforced soil walls assessed against the results of meticulously constructed and completely instrumented full-scale models are a complementary approach for gathering adequate data to assess the validity of present models or calibrate new design techniques. The synthetic data generated by validated numerical models can then be used to fill knowledge gaps in the available database of physical measurements [37,38].

An additional benefit of validated numerical models is that they can be used to estimate wall deformations, which are required for performance-based structural design [1,35]. The behavior of reinforced MSE wall has also been investigated using three-dimensional (3D) finite element analysis. Ref. [39] studied three 0.8 m high reinforced earth model walls with strip footing surcharge at the wall facing which were subjected to experimental and numerical analysis. The research aims to discover how the strength and geometry of wire mesh affect the failure mechanism. The influence of sidewall friction (3D model) and reinforcing stiffness and strength (2D model) on the failure mechanism of the walls was also studied using numerical modeling. Independent tests and results were used to derive the parameters for the numerical models, which were then compared to the experimental findings. Ref. [38] evaluated the stability analysis of road embankment by using finite element software Plaxis 3D. Two-basic modeling was investigated in this paper. First, the performance of a road embankment without any reinforcing material was investigated. The second model of embankment was investigated using geocells with various height to width ratios. The raft foundation for the road embankment was created by the small settlement and deflected shape of the geocell mattress. The results of this investigation revealed that the tensile strength of the geocell had a significant impact on embankment stability.

The primary aim of this research was to create a numerical model that could accurately match the measured performance characteristics of a large-scale MSE wall reinforced with deformed steel bars embedded in tire shred-sand mixture. The necessity to construct a numerical model that can generate complementary synthetic data for the MSE wall with deformed steel bars with tire shred-sand mixture as backfill material was the driving reason behind the effort [40,41]. Plaxis 3D, a software, was utilized to create the numerical model.

## 2. Methods

The work methods are divided into two main sections: experimental investigations and numerical analysis.

### 2.1. Experimental Work

#### 2.1.1. Materials Used

The materials used in the experimental work are divided into three material types: The Investigated Sand, Tire shreds and the Deformed steel bars.

#### The Investigated Sand

Locally accessible, poorly-graded sandy soil (SP) that adheres to the Unified Soil Classification System (USCS) was employed as a significant filler. The particle size distribution was determined using the standard soil particle size analysis testing method [42]. Figure 1 illustrates the particle size distribution of the sand used in this study. The coefficients of uniformity and curvature were calculated to be 1.75 and 1.37, respectively, for a factor of 3.2. The fineness module for sand was computed as 2.73, according to ASTM C33 [43].

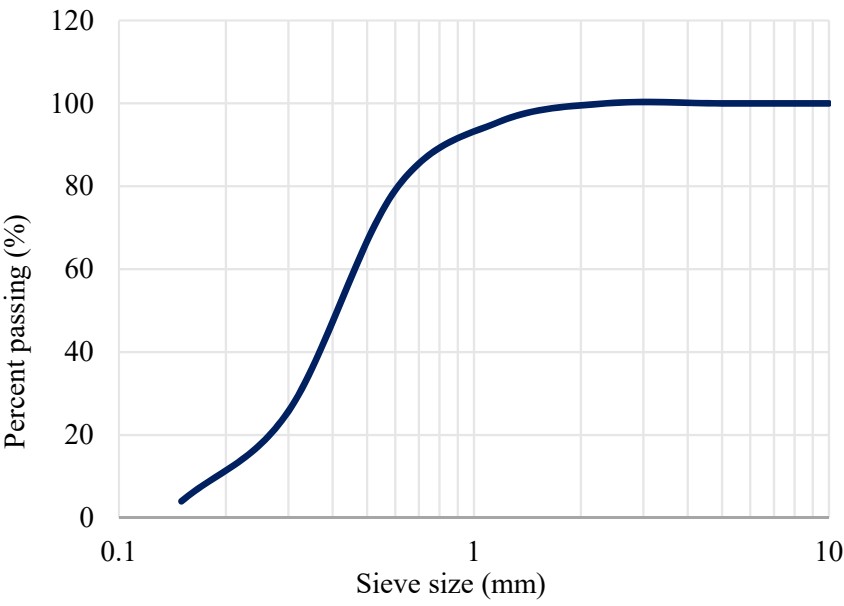

**Figure 1.** Grains' size distribution curve.

#### Tire Shreds

Tire shreds with lengths of 50 mm were used in this study. The 50 mm tire shreds had the exact dimensions. Tires were chopped to the proper sizes according to ASTM D-6270 [44]. The tire shred length was calculated at random using measuring tape and the geometric criterion [24]. Tire shreds have a low unit weight (varying from 2.4 kN/m$^3$ to 7.0 kN/m$^3$), high hydraulic conductivity (ranging from $1.8 \times 10^{-3}$ cm/s to 15.4 cm/s), and reasonably good shear strength, which make them ideal for civil engineering applications. Tire shreds without steel wires are lighter than water, and hence the specific gravity GS of tire shreds from 50 mm to 100 mm in length is 1.23 [45–47]. In Figure 2, the visual aspect of tire shreds of 50 mm size are depicted.

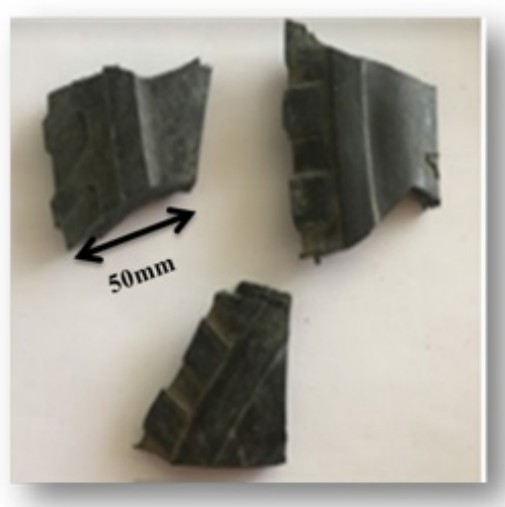

**Figure 2.** Visual aspect of tire shreds of 50 mm.

### 2.1.2. Deformed Steel Bars

Deformed round steel bars with a tensile strength of 276 N/mm$^2$ were chosen as reinforcements, as shown in Figure 3. The diameter was kept at least three times bigger than the average (D50) particle size to achieve appropriate contact friction [23,30]. Thus, the diameter of the deformed bars utilized in this examination was 12.7 mm (No. 4), and the bar length was kept at 0.7 H (427 mm).

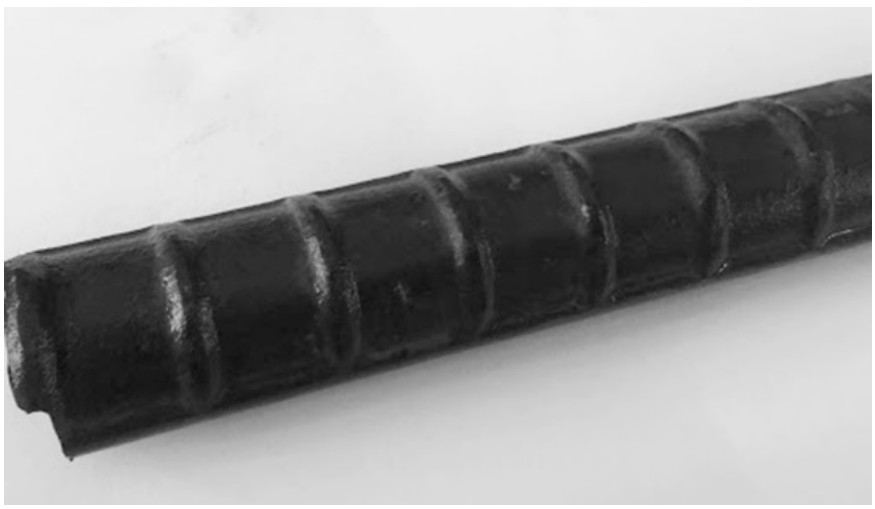

**Figure 3.** Visual aspect of deformed steel bar.

### 2.1.3. Physical Setup and Test Procedure

The fabrication of a large box with dimensions of 0.9 × 1.2 × 1.5 m was carried out for experimental purposes using steel material with a thickness of 2 mm, as illustrated in Figure 4, which shows that the box additionally has a cross and vertical strips to reinforce it, and prevent it from bulging during load application. After the box has been prepared, it is installed in the material testing laboratory underneath the loading frame, and a normal load is applied using a hydraulic jack with a capacity of 200 kN. Two types of backfill material were utilized. One sample was sand, and the other was a tire shred-sand mixture. Both samples were reinforced with deformed steel bars. The reinforcement was placed in nine layers, with the center to center spacing of 0.15 m. The ratio of tire shreds to the sand mixture was set at 20/80 (tire shred-sand) by weight, and the second medium consisted solely of reinforced sand. The wall was then instrumented with potentiometers in both

vertical and horizontal positions. Horizontal potentiometers were installed to record the wall's lateral face deflection, while vertical potentiometers were employed to determine the medium's settlement concerning the application of normal load.

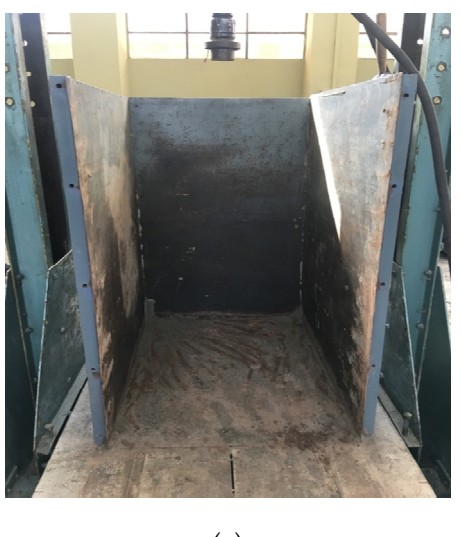 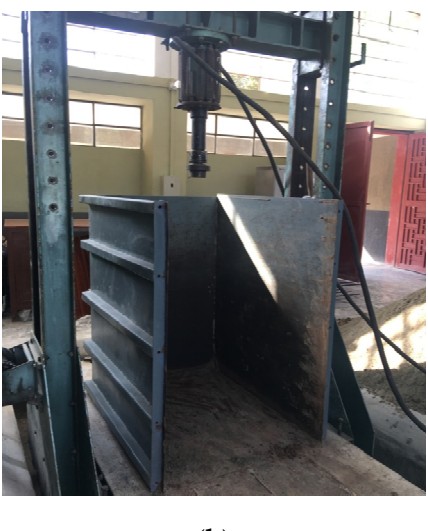

(**a**)                                                                                                              (**b**)

**Figure 4.** (**a**) The full scale model of MSE wall; (**b**) The MSE model with loading frame and hydraulic jack.

Moreover, load cells were installed at the point of vertical load application for continuous load measurement. A data logger with software configuration was installed for data monitoring, display, and storage throughout the experiment. While removing the front half of the box, a thin piece of plywood was affixed throughout the depth to prevent face movement. Three plates of 0.61 m, 0.45 m, and 0.25 m were positioned above the top plate of the box for load application to distribute the uniform load throughout the sample. The size of the tire-derived aggregate was a significant consideration factor because the size of the tire shreds selected will affect all the backfill material's properties, including compaction, shear strength, and density. To compact layers rapidly and effectively with minimal effort or mechanical methods, the maximum size of tire shreds was selected at 50 mm. Moreover, the size of the shreds has a significant impact on the sand sample consistency. Furthermore, the 50 mm size yielded highly accurate and reliable data.

2.1.4. Backfilling of MSE Wall

Before performing the test, it is critical to reduce the friction between the sidewalls and the soil. A layer of thin, greased polyethylene was employed to reduce sidewall friction to achieve this. For testing, the soil was compacted in 20 layers, each 90 mm thick, using a hand tamper with a $300 \times 300$ mm size and a weight of 0.10 kN. One of the most important aspects of sample preparation was the compaction procedure, which ensured that each layer had the same density. The soil layers were compacted with three blows of a 0.10 kN steel tamper dropped from a height of 250 mm, using a revolutionary center-to-center compaction cycle. The same procedure was then employed for all the samples. The compacted sample is represented in Figure 5a.

The ratio was 20/80, which meant that 20% of the tires and 80% of the sand were utterly mixed and filled as a backfill behind the box. A deformed bar of 12.7 mm diameter was a reinforcement with a horizontal and vertical spacing of 0.15 mm. As a result, an arrangement of nine layers was built up to the top, accommodating nine bars in a single column. The box contained 45 bars. The embedding length was limited to 0.7 times the wall's height, according to FHWA manual [8]. After the filling procedure was completed, a top plate with a smooth and leveled surface was placed above it to distribute vertical force when applied to achieve consistent stresses throughout the medium, ensuring a

reliable result and avoiding discrepancies. A large-type straining frame with a capacity of 200 kN and control application of load was utilized for load application. When the model failed, the medium was removed, and the procedure was repeated using only sand with reinforcement. The model setup is depicted in Figure 5b, below, with the bars arranged and the material compacted.

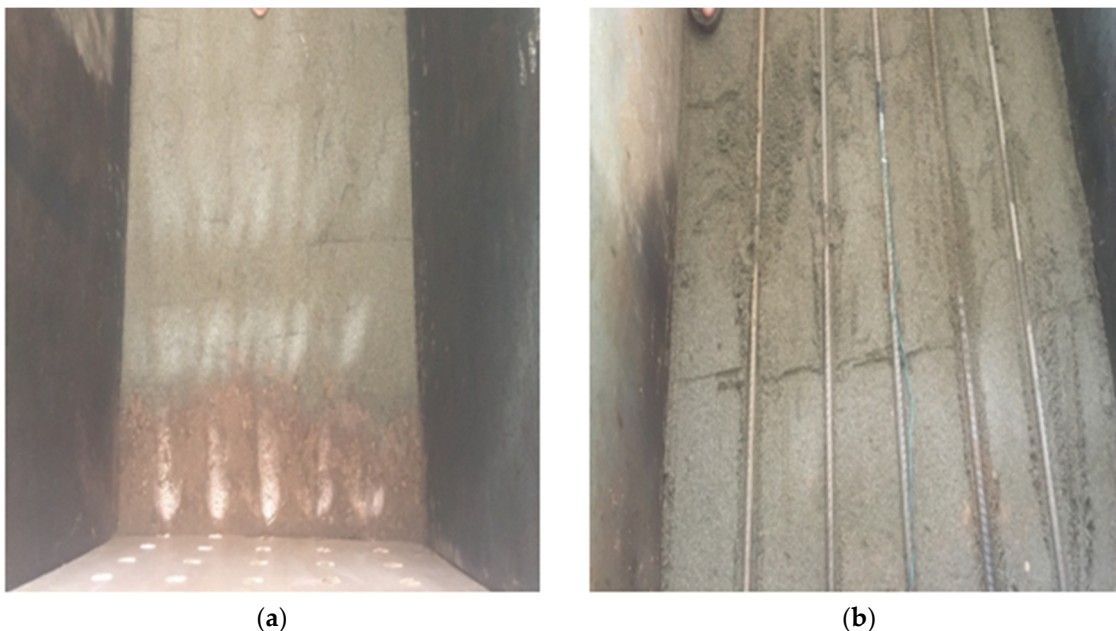

(**a**)        (**b**)

**Figure 5.** (**a**) The compacted sample of sand (**b**) MSE wall with strip arrangement.

2.1.5. Instrumentation of the Model

Various instruments, including potentiometers [32] in vertical and horizontal locations, and load cells for monitoring the progressive increase of force, have been installed to observe face deflection at facing and soil settlement due to the application of load. However, each measuring instrument was calibrated before installation to ensure accurate measurements. Figure 6a depicts the finalized box assembly and the loading frame along with the installation of numerous instrumentation. In front of the face, a ladder was erected to support and position lateral deflection gauges. A plywood piece was added to the front face to support the face and provide a thick and strong medium for gauge compression of the top flat plate as well as the circular plate. Figure 6b represents the schematic view of the complete model. The box was securely placed in the center of the straining frame to bring the loading point exactly in the middle part. This will result in equal load distribution and uniform settlement on both sides. All of the instruments were connected to a data logger for data collection. Moreover, the Datalogger was linked to a computer with the MATLAB software package to view data and observe continuous settlement, deflection, and incremental loads.

*2.2. Numerical Modeling*

Numerical modeling is consistently recognized as a vital step in validating experimental methods and steps. Geotechnical software Plaxis 3D v.21 ultimate was used in this circumstance due to its accuracy in results and user-friendly interface. Plaxis 3D, a numerical software application, was used to precisely model and appraise the setup, depending exclusively on geotechnical responses. While commencing to model numerically, all types of data needed to incorporate in software were determined and collected. Various parameters' provision and inclusion were critical since they influenced results to a greater extent [38,40,48]. As a result, the requisite tests, such as the direct shear test for shear

strength parameters and other tests such as compaction test, which were performed to calculate the maximum dry density that we desired, were carried out at the same time.

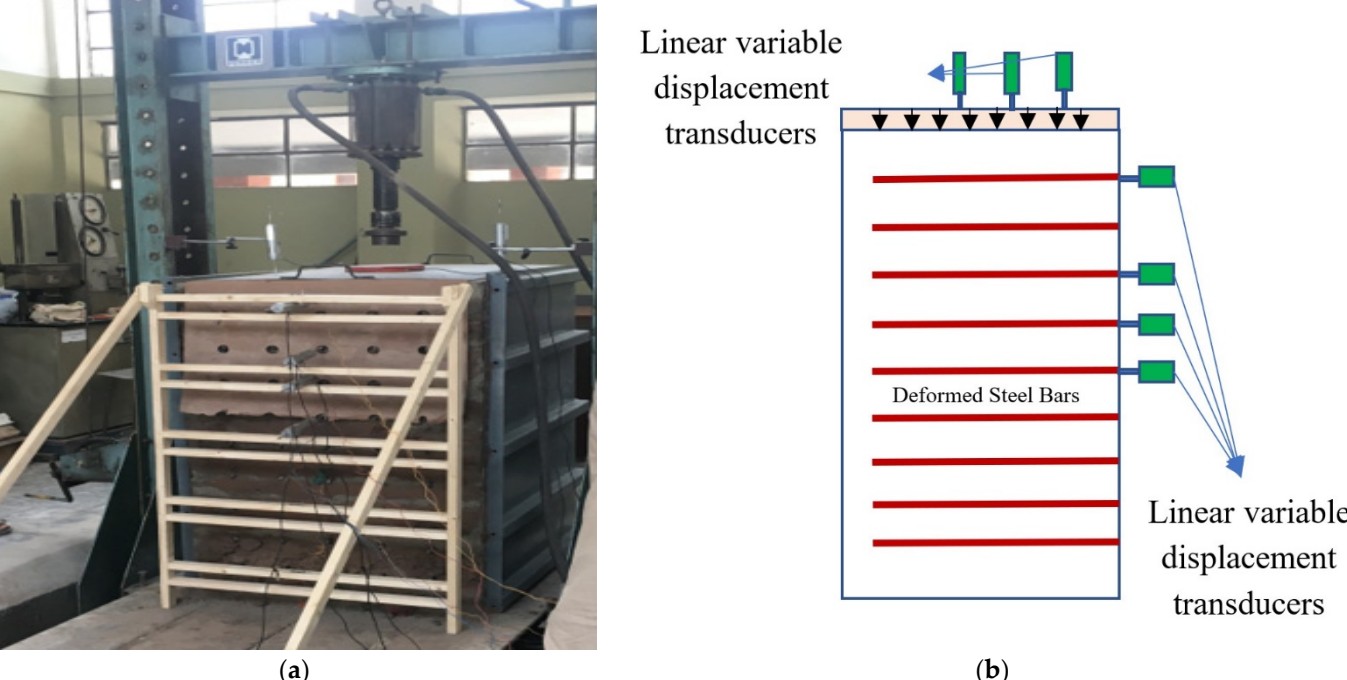

**Figure 6.** (**a**) Large scale model of MSE along with instrumentation, loading plates, and hydraulic jack; (**b**) Schematic view of MSE wall.

Plaxis 3D is a well-known and user-friendly geotechnical engineering program. Most academic scholars use Plaxis 3D in the foundation and geotechnical numerical simulation since it is solely based on geotechnical [4,49]. Plaxis 3D can model complex engineering designs and geotechnical structures such as tunnels, foundations, and embankments [14,36]. Furthermore, the software could analyze a structure in both static and dynamic modes. As a result, obtaining well-established data prior to commencing modeling in Plaxis is critical [38,50]. Special procedures and properties for defining hydrostatic and non-hydrostatic pressure were incorporated into software [51]. Plaxis software is known for considering interaction features while examining things such as soil-structure interaction, which is vital. It can depict a wide range of elements, including plates, piles, geogrids, soil, and interfaces, among others [51,52].

The Plaxis has a very smooth flow of constructing the model with sequential steps, as shown in Figure 7. The following are the detailed procedures of constructing a model [50].

The numerical modeling was performed using an official Plaxis 3D v.21 Ultimate License at Saint Petersburg Polytechnic University, Russian Federation, subscription type: CNTI-SPbPU (1006650066).

2.2.1. The Parameters of the Model

Plaxis 3D provides exceptionally reliable steps for the construction of the MSE wall. As a result, all three sides of the box and the bottom surface were represented in this case in the form of a plate [53]. The front face of the box was removed. A flat surface was constructed as a plate element after the box was created [53] to place it above of medium so that it would distribute the load equally. The box was then filled with two types of medium, each having the appropriate attributes listed in Table 1 for each circumstance. The bars were modeled as piles with an embedded length of 0.7 times the wall's height, as per the specification [6]. Both horizontally and vertically, the center-to-center space between bars was fixed at 0.15 m. All boundary conditions were applied with precision. In the last

stage, a normal load of 20 kPa was applied at first, then gradually increased to 40 kPa in increments. Figure 8 demonstrates the model in detail.

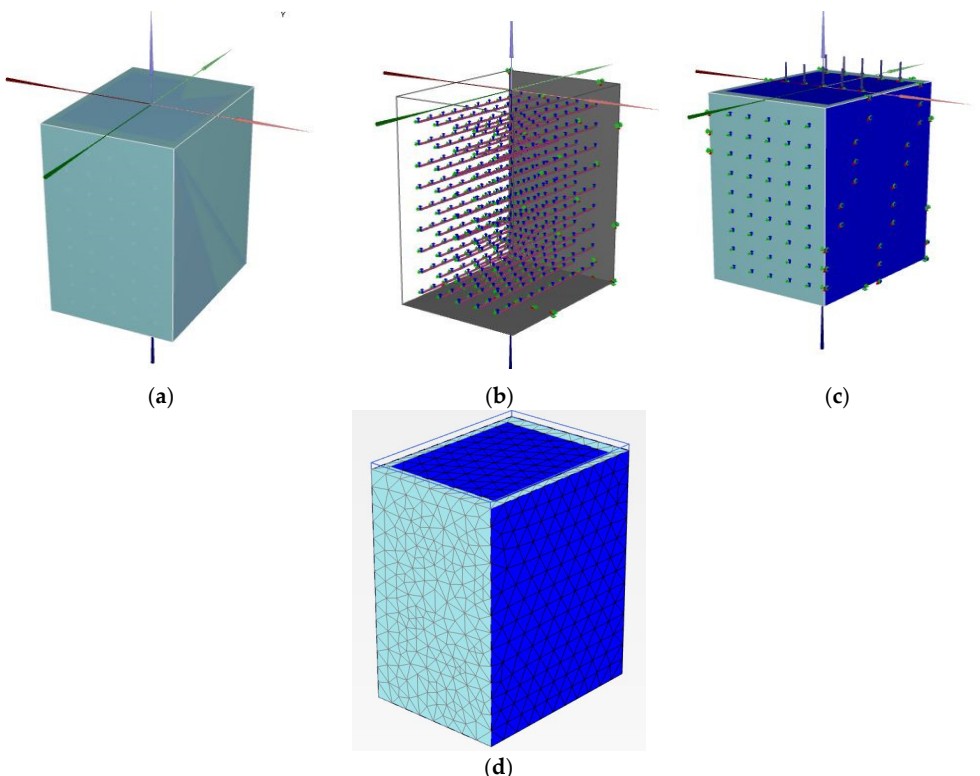

(a)　　　　　　　　　(b)　　　　　　　　　(c)

(d)

**Figure 7.** Different stages of the model construction phase in Plaxis 3D: (**a**) Soil phase; (**b**) Structure phase (model with box and deformed bars); (**c**) Illustration of the top plate with load application; (**d**) Meshed model of Plaxis 3D.

**Table 1.** Properties of material and specifications.

| Material | Description | Type | Value |
|---|---|---|---|
| Tire Shred-Sand Mixture | Unit Weight, $Y_w$ (kN/m³) | Unsaturated | 19 |
| | Unit Weight, $Y_w$ (kN/m³) | Saturated | 19 |
| | Poisson Ratio, $\upsilon$ | - | 0.35 |
| | e | Initial | 0.5 |
| | E (MPa) | - | 7.143 |
| | c (kN/m²) | - | 0.54 |
| | Rint | - | 0.67 |
| | Phi (Degree) | - | 39 |
| Only Sand | Unit Weight, $Y_w$ (kN/m³) | Unsaturated | 23 |
| | Unit Weight, $Y_w$ (kN/m³) | Saturated | 23 |
| | Poisson Ratio, $\upsilon$ | - | 0.35 |
| | e | Initial | 0.5 |
| | E (MPa) | - | 10 |
| | c (kN/m²) | - | 0.1 |
| | Rint | - | 0.67 |
| | Phi (Degree) | - | 32 |

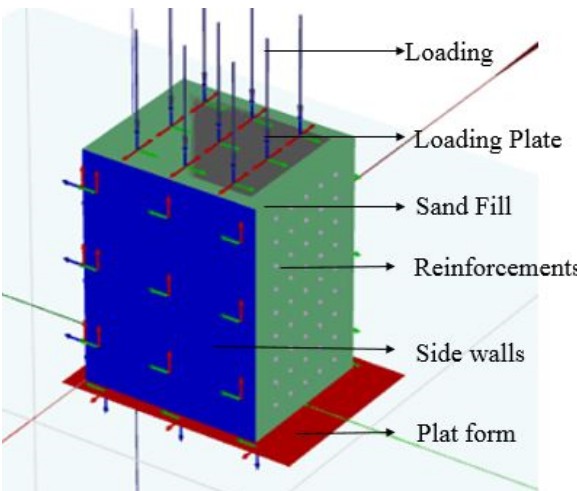

**Figure 8.** Three dimensional numerical models with description.

2.2.2. Material Properties and Specifications

There were two sorts of materials used. The first was made entirely of sand with reinforcement, while the second was constructed of a tire shred-sand mixture. According to the specifications, all types of attributes for both mediums were correctly specified from experimental test results. The backfill medium was specified using Mohr–Columb criterion with drained circumstances in both cases [39,52]. The interface property between soil and rebars was observed to be 0.67. Table 1 shows different soil properties.

The reinforcing bars designed as embedded piles have the following properties described in Table 2.

**Table 2.** Reinforcement material and box properties.

| Deformed Steel Bars Act as Embedded Piles | |
|---|---|
| E (GPa) | 200 |
| Unit weight (Y), kN/m$^3$ | 78.5 |
| Diameter (m) | 0.01 |
| $T_{top\ max}$ (kN/m) | 1 |
| $T_{top\ Bottom}$ (kN/m) | 1 |
| Vertical Spacing, $S_v$ (m) | 0.15 |
| Horizontal Spacing, $S_h$ (m) | 0.15 |

The box was designed as a plate element having the following properties (Table 3).

**Table 3.** The properties of the box.

| Box Properties | |
|---|---|
| E (GPa) | 200 |
| Unit weight (Y), kN/m$^3$ | 78.5 |
| Thickness (mm) | 2 |

## 3. Results and Discussion

### 3.1. Experimental Model Results

The comparative study of tire shred mix with reinforcement and only sand with reinforcement was carried out to analyze the performance of shreds against the only sand medium.

Figure 9 clearly indicates that at 20 kPa, both cases deflected approximately equally, with the tire shred-sand mixture with reinforcement deflecting 16 percent more. In the other two incremental loads, however, the deflection of sand was larger than that of tire shred-sand combined with the reinforcement. Furthermore, the tire shred-sand mixture with the reinforcement is lightweight, reducing overburden pressure and less face deflection than sand alone. The face deflected due to the higher density and greater overburden pressure induced by the sand. The sand deflection was 35 percent higher at 30 kPa than tire shred-sand with reinforcement, but the difference was 57 percent at 40 kPa, which is considerable. According to the literature, a 50% to 60% discrepancy should be detected for sand with a reinforcement case. This finding was in agreement with the results obtained by [54]. According to Figure 10, a combination of 20 kPa and 30 kPa results in a 33 percent increase in settlement, but this difference rises to 36 percent at 40 percent. The sand has a low void ratio compared to the tire shred-sand mixture. It means that the sand had a higher density than the other combination. Their compressibility was consistently lower than that of the tire shred-sand-reinforcement combination, with a difference of 50 to 60%.

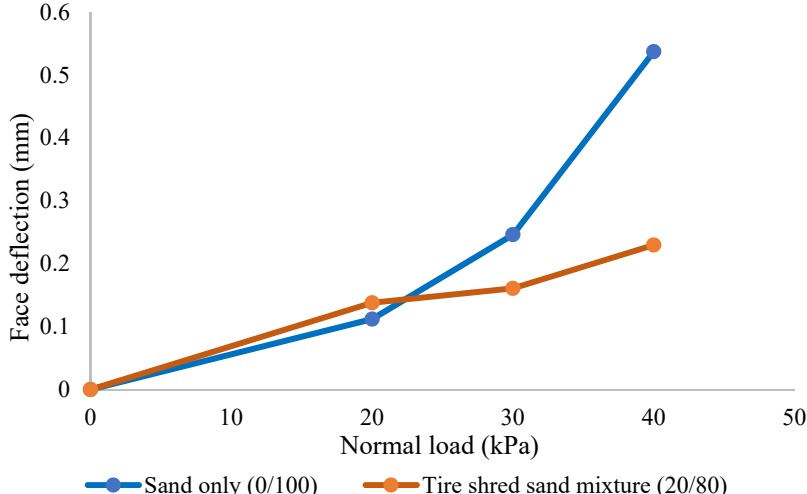

**Figure 9.** Comparison of wall displacement profiles for different surcharge pressures on tire shred-sand mixtures (20/80) and sand only (0/100).

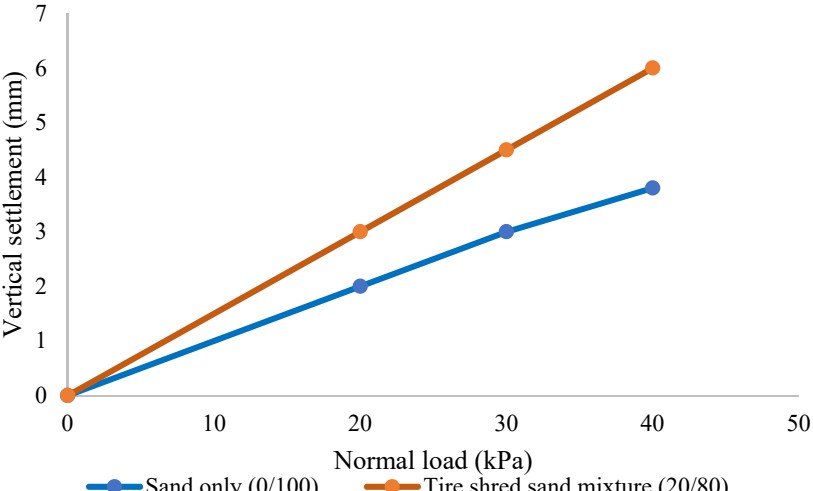

**Figure 10.** Comparison of vertical settlement with surcharge pressures on tire shred-sand backfill (20/80) and sand only (0/100).

Tire shred is a material that is elastic and compressible [27]. As a result, it compresses to a greater extent as the load increases. A comparison of all of these numbers for two types of medium is shown in Table 4.

### 3.2. Numerical Model Results

Plaxis 3D software was employed for numerical modeling. All types of components were properly modeled and allocated properties. Deformed bars were represented as embedded piles, and both forms of media, i.e., only sand with reinforcement and tire shred-sand mixture with the reinforcement, were modeled as Mohr–Columb criterion. Incorporating material parameters is critical for accuracy and precision in findings. Similarly, applying boundary conditions is quite essential to assign to the model. The failure planes for both the mediums are shown in Figure 11.

**Table 4.** Face deflection and max settlement for tire shred-sand mixture with reinforcement and for sand and reinforcement only.

| Load (kPa) | Descriptive Case | Max. Top Face Deflection (mm) | Max. Settlement (mm) |
| --- | --- | --- | --- |
| 0 | | 0 | 0 |
| 20 | Sand with Reinforcement | 0.112 | 2 |
| 30 | | 0.246 | 3.0 |
| 40 | | 0.537 | 3.8 |
| 0 | | 0 | 0 |
| 20 | Tire Shred-sand Mixture with reinforcement | 0.138 | 3 |
| 30 | | 0.161 | 4.50 |
| 40 | | 0.230 | 6 |

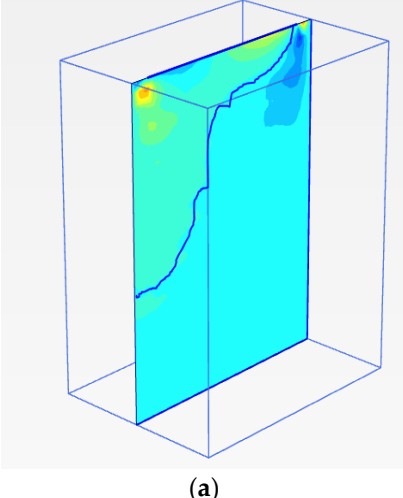 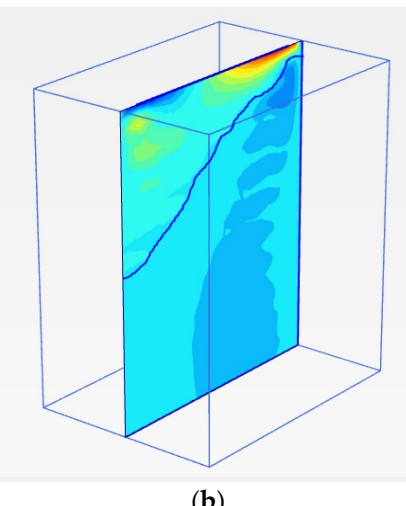

(**a**)             (**b**)

**Figure 11.** (**a**) Major strains observed by digital image correlation analysis in Plaxis 3D at the sliding surface in the sand only; (**b**) Major strains observed by digital image correlation analysis in Plaxis 3D at the sliding surface of tire shred-sand mixture.

According to the comparative evaluation shown in Figure 12, the deflection is relatively the same at 20 kPa, but at the other two loads, 30 kPa and 40 kPa, the tire shred-sand mixture with reinforcement gave a 36 percent and a 58 percent reduction in deflection, respectively, when compared to sand with the reinforcement. This is due to the lightweight qualities of the combination, which result in decreased deflection due to lower overburden pressure.

This instance was also supported by the literature, which states that the percent difference should be between 50 and 60 percent.

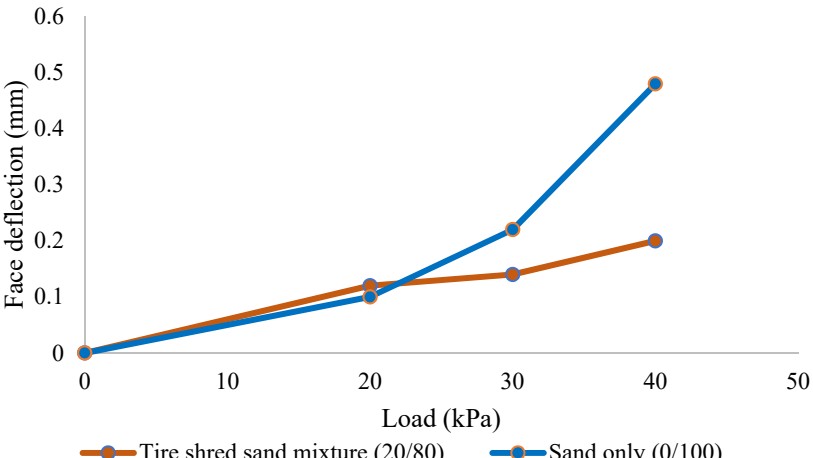

**Figure 12.** Comparison of face deflection for sand only (0/100) and tire shred-sand mixture (20/80) in Plaxis 3D.

The sand has a minimum void ratio compared to the tire shred-sand mixture. Ultimately, it means that the density of sand is higher than the mixture. Due to increased density, their compressibility is always lower than tire shred-sand mixture with reinforcement [55]. Secondly, the tire chips are elastic and compressible material. Therefore, with increased load, it compresses up to a greater extent. It has been evaluated that at 20 kPa and 30 kPa, 40% and 36% respective rise in the settlement was noted for a mixture, but at 40 kPa, this difference lowers to 33%, as shown in Figure 13. Table 5 below compares all these values for two types of medium.

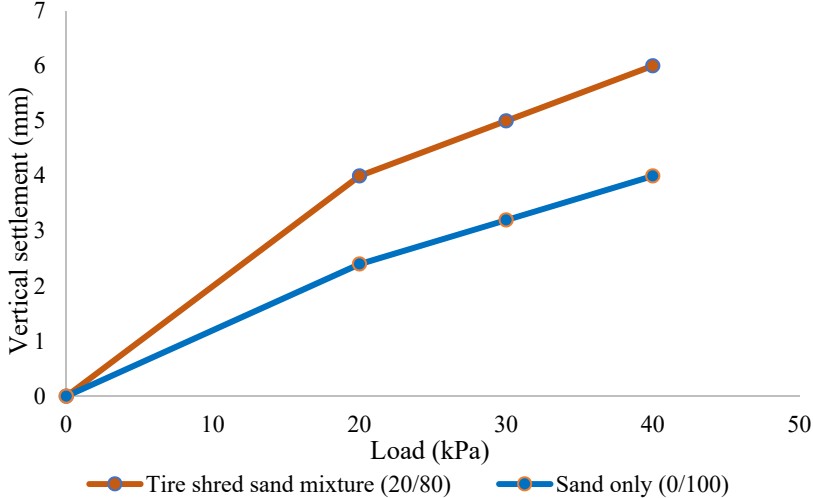

**Figure 13.** Comparative graph of vertical settlement for sand only (0/100) and tire shred-sand mixture (20/80) in Plaxis 3D.

### 3.3. Comparison between Numerical and Experimental Model

In the case of the experimental study, the maximum load supported by both types of the medium was 40 kPa, which results in the failure of the model with defined slip planes. At this stage, all settlement and deflection values were noted for every increment. However, in the numerical study, the wall was observed till the load to 300 kPa for tire shred-sand mixture with reinforcement and 100 kPa for sand with reinforcement. After these loads, the models failed, and further analysis was not possible. In the case of sand, the percent

difference between numerical and experimental values was only 12%. Experimental values observed were 12% higher than that of numerical. However, in the case of tire shreds, this difference seems to be 15% for the same conditions. In terms of the settlement, the difference for sand was found to be 5%, which is in the acceptable range for a peak load of 40 kPa but 0% in case of a tire, shreds for the same load and conditions. The comparative analysis between experimental and numerical models is described in Table 6.

**Table 5.** Comparative values of tire shred-sand mixture with reinforcement and only sand with reinforcement from Plaxis 3D.

| Description | Load (kPa) | Deflection (mm) | Settlement (mm) |
|---|---|---|---|
| | 0 | 0 | 0 |
| | 20 | 0.12 | 4 |
| | 30 | 0.14 | 5 |
| | 40 | 0.2 | 6 |
| Tire shred-sand mix with reinforcement | 60 | 0.36 | 8.5 |
| | 80 | 0.48 | 11 |
| | 100 | 0.9 | 13 |
| | 200 | 2.4 | 24 |
| | 300 | 4.8 | 34 |
| | 300 and above | fails | fails |
| | 0 | 0 | 0 |
| | 20 | 0.1 | 2.4 |
| Only sand with reinforcement | 30 | 0.22 | 3.2 |
| | 40 | 0.48 | 4 |
| | 60 | 1.6 | 6 |
| | 80 | 2.75 | 9 |
| | 100 and above | fails | fails |

**Table 6.** Comparison of numerical and experimental models.

| Description | Loads (kPa) | Face Deflection (mm) | | Vertical Settlement (mm) | |
|---|---|---|---|---|---|
| | | Numerical | Experimental | Numerical | Experimental |
| | 0 | 0 | 0 | 0 | 0 |
| | 20 | 0.12 | 0.138 | 4 | 3 |
| | 30 | 0.14 | 0.161 | 5 | 4.5 |
| | 40 | 0.2 | 0.23 | 6 | 6 |
| Tire shred-sand mix with reinforcement | 60 | 0.36 | - | 8.5 | - |
| | 80 | 0.48 | - | 11 | - |
| | 100 | 0.9 | - | 13 | - |
| | 200 | 2.4 | - | 24 | - |
| | 300 | 4.8 | - | 34 | - |
| | 300 and above | | fails | | |
| | 0 | 0 | 0 | 0 | 0 |
| | 20 | 0.1 | 0.112 | 2.4 | 2 |
| | 30 | 0.22 | 0.246 | 3.2 | 3 |
| Only sand with reinforcement | 40 | 0.48 | 0.537 | 4 | 3.8 |
| | 60 | 1.6 | - | 6 | - |
| | 80 | 2.75 | - | 9 | - |
| | 100 and above | fails | - | fails | - |

## 4. Conclusions and Future Prospects

According to the experimental and finite element analysis investigation results, the following are concluded:

1.  From the results of the large-scale model, the face deflection observed at 20 kPa observed in the sand only and in tire-shred, the sand mixture is relatively the same; at 30 kPa and 40 kPa, the tire shred-sand mixture with reinforcement gave 36% and 58% reduction in deflection as compared to sand with the reinforcement. This is because of the lightweight properties of the mixture, which results in less overburden pressure resulting in lower deflection.
2.  The vertical settlement observed in the large-scale model, the tire shred-sand mixture at 20 kPa and 30 kPa, shows higher settlement than sand, i.e., a 33% rise in the settlement was noted. At 40%, this difference rises to 36% due to the sand minimum void ratio compared to tire shred-sand mixture, and due to increased density, their compressibility was consistently lower than that of tire shred-sand mixture with reinforcement.
3.  The experimental values observed were 12% higher than that of numerical. In the case of sand, the percent difference between numerical and experimental values is only 12%. In the case of tire shreds, this difference seems to be 15% for the same conditions.
4.  For vertical settlement in experimental and numerical values, the difference for sand was found to be 5%, which is in the acceptable range for a peak load of 40 kPa but 0% in case of a tire, shreds for the same load and conditions.

The large-scale wall used in this study demonstrates the effect of backfill material reinforced with deformed steel bars, wall construction details, and measurements needed to calibrate a matching numerical model utilizing program Plaxis 3D. Numerical models that have been calibrated, can be used to optimize the design of reinforced soil retaining wall structures during the design stage, to check expected loads and to undertake parametric analyses. Synthetic data derived from parametric studies may also be used to fill in knowledge gaps for wall structures.

**Author Contributions:** Conceptualization, B.J.K., M.A. and M.M.S.S.; methodology, B.J.K., M.A. and M.M.S.S.; software, B.J.K. and M.M.S.S.; formal analysis, B.Z., M.N. and I.A.; resources, M.M.S.S.; data curation, B.J.K., M.N. and M.A.; writing—original draft, B.J.K., M.A., M.M.S.S., B.Z. and I.A.; writing—review and editing, B.J.K., M.M.S.S. and M.A.; supervision, B.J.K. and M.M.S.S.; funding acquisition, M.M.S.S. All authors have read and agreed to the published version of the manuscript.

**Funding:** This research is partially funded by the Ministry of Science and Higher Education of the Russian Federation under the strategic academic leadership program 'Priority 2030' (Agreement 075-15-2021-1333 dated 30 September 2021).

**Institutional Review Board Statement:** Not applicable.

**Informed Consent Statement:** Not applicable.

**Data Availability Statement:** The data used to support the findings of this study are included in the article.

**Conflicts of Interest:** The authors declare that they have no known competing financial interests or personal relationships that could have appeared to influence the work reported in this paper.

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
