# Peer review of "Experimental and Numerical Evaluation of Mechanically Stabilized Earth Wall with Deformed Steel Bars Embedded in Tire Shred-Sand Mixture"

_buildings, doi:10.3390/buildings12050548_

Round 1

Reviewer 1 Report

see the file attached

Reviewer 2 Report

The introduction provide sufficient background and include all relevant references,and the research design is appropriate.What's more, the methods are adequately described and the results are clearly presented, so the conclusions are supported by the results.
In my opinion,it can be accepted in present form.

Reviewer 3 Report

This paper presents the experimental and numerical analysis behavior on Mechanically Stabilized Earth Wall (MSE). The overall quality of is OK but the tests results are not well-presented. The numerical study seems insufficient and needs to be further supplemented.

  1. The introduction is so length, the author may decompose one paragraph into several and group the past research according to its respective topic.
  2. 5 (a) and (b) are not clear.
  3. According to conclusion 3, the numerical does not generate close prediction to the test result. In the case, how to demonstrate the accuracy of the numerical model?
  4. Normally, there should be parametric study to further evaluate the effect of each parameter after the validation and calibration of numerical model.

Round 2

Reviewer 3 Report

have revised according to comments